# Application of Standardized Antimicrobial Administration Ratio as a Motivational Tool within a Multi-Hospital Healthcare System

**DOI:** 10.3390/pharmacy9010032

**Published:** 2021-02-07

**Authors:** Stephanie Shealy, Joseph Kohn, Emily Yongue, Casey Troficanto, P. Brandon Bookstaver, Julie Ann Justo, Hana R. Winders, Sangita Dash, Majdi N. Al-Hasan

**Affiliations:** 1Department of Clinical Pharmacy and Outcomes Sciences, University of South Carolina College of Pharmacy, Columbia, SC 29208, USA; Stephanie.Shealy@imail.org (S.S.); bookstaver@cop.sc.edu (P.B.B.); justoj@cop.sc.edu (J.A.J.); hwinders@cop.sc.edu (H.R.W.); 2Department of Clinical Pharmacy, Prisma Health–Midlands, Columbia, SC 29203, USA; joseph.kohn@prismahealth.org (J.K.); Emily.Yongue@prismahealth.org (E.Y.); Casey.Troficanto@prismahealth.org (C.T.); 3Department of Medicine, Division of Infectious Diseases, University of South Carolina School of Medicine, Columbia, SC 29209, USA; Sangita.Dash@uscmed.sc.edu; 4Department of Medicine, Division of Infectious Diseases, Prisma Health–Midlands, Columbia, SC 29203, USA

**Keywords:** antibiotics, antimicrobial use, antimicrobial stewardship, metrics

## Abstract

The standardized antimicrobial administration ratio (SAAR) is a novel antimicrobial stewardship metric that compares actual to expected antimicrobial use (AU). This prospective cohort study examines the utility of SAAR reporting and inter-facility comparisons as a motivational tool to improve overall and broad-spectrum AU within a three-hospital healthcare system. Transparent inter-facility comparisons were deployed during system-wide antimicrobial stewardship meetings beginning in October 2017. Stakeholders were advised to interpret the results to foster competition and incorporate SAAR data into focused antimicrobial stewardship interventions. Student’s *t*-test was used to compare mean SAARs in the pre- (July 2017 through October 2017) and post-intervention periods (November 2017 through June 2019). The mean pre-intervention SAARs for hospitals A, B, and C were 0.69, 1.09, and 0.60, respectively. Hospital B experienced significant reductions in SAAR for overall AU (from 1.09 to 0.83; *p* < 0.001), broad-spectrum antimicrobials used for hospital-onset infections (from 1.36 to 0.81; *p* < 0.001), and agents used for resistant gram-positive infections in the intensive care units (from 1.27 to 0.72; *p* < 0.001) after the interventions. The alignment of the SAAR across the health-system and sustained reduction in overall and broad-spectrum AU through implementation of inter-facility comparisons demonstrate the utility in the motivational application of this antimicrobial use metric.

## 1. Introduction

Antimicrobial stewardship programs (ASPs) have long sought a standardized method for benchmarking antimicrobial use, allowing comparative antimicrobial stewardship metrics between hospitals and assessment of the effectiveness of targeted interventions [1,2,3]. Traditional facility-specific antimicrobial use metrics include days of therapy, defined daily dose, and antimicrobial cost [4]. Large healthcare systems may be able to internally develop a robust approach to antimicrobial use benchmarking [5]. However, the utility of these tools may be limited in smaller healthcare systems and community hospitals. In response to this need, the Centers for Disease Control and Prevention (CDC) National Healthcare Safety Network (NHSN) introduced the standardized antimicrobial administration ratio (SAAR) as a metric included in the antimicrobial use option [6]. The primary objective of the antimicrobial use option is to facilitate risk-adjusted inter- and intra-facility antimicrobial use benchmarking using facility-reported data and the provision of a facility-specific SAAR. The SAAR is a novel, NHSN-developed antimicrobial stewardship metric that compares observed to predicted antimicrobial use. The SAAR was first provided to participants in the antimicrobial use option in 2015. Initially based on models developed and applied to nationally aggregated 2014 adult and pediatric antimicrobial use data, the models used to provide the SAAR have since been updated, using 2017 adult and pediatric data [7]. The SAAR is provided for overall antimicrobial use and specific patient care locations and antimicrobial categories. An SAAR statistically greater than one indicates more antimicrobial use than expected, while an SAAR statistically less than one indicates less antimicrobial use than expected. When interpreting the SAAR, facilities should be aware of its limitations. The models used currently do not reflect expected antimicrobial use based on the case-mix index or infectious disease burden [7]. In addition, the data provided are highly dependent on the accuracy of the facility-reported data. The NHSN provides resources for facilities in order to educate and encourage internal validation of data submitted to the antimicrobial use option. Lastly, facilities should be aware of potential seasonal variation in the SAAR [8]. Despite these limitations, the SAAR is considered the most direct metric for the measurement of antimicrobial stewardship performance [9]. To encourage antimicrobial use option participation, the CDC specifies tracking of antimicrobial use and reporting to NHSN as a priority to achieve the core elements of hospital antibiotic stewardship programs [10].

The value of antimicrobial use participation for ASPs has been previously demonstrated [11,12,13,14,15,16]. The present literature describes the utility of the SAAR as an identifier of outlier patient care areas and antimicrobial categories requiring additional ASP support and as a metric to measure the impact of targeted interventions [11,12,13,14]. The impact of transparent distribution of the SAAR and inter-facility comparisons within a multi-hospital health-system to motivate local providers and ASP leads has not been previously described. This before and after prospective cohort study evaluates the impact of the SAAR as a motivational tool on improving overall and broad-spectrum antimicrobial use across a three-hospital healthcare system.

## 2. Materials and Methods 

### 2.1. Setting 

This study was conducted at three Prisma Health—Midlands hospitals in South Carolina, USA that varied in bed capacity and patients’ characteristics. These hospitals provide a variety of medical, surgical, and subspecialty services to patients in the Midlands of South Carolina. The ASP consists of an infectious diseases physician and pharmacist leads in each hospital who meet regularly to discuss issues related to antimicrobial use as well as develop and sustain focused antimicrobial stewardship interventions. Participation of Prisma Health—Midlands in NHSN’s antimicrobial use option began in July 2017. Prior to participating in the antimicrobial use option, the ASP used locally generated metrics, such as days of therapy, to deploy targeted interventions to optimize antimicrobial use through syndrome-specific and prospective audit and feedback interventions [17].

### 2.2. Study Design and Definitions 

This before and after prospective cohort study examines the impact of the motivational application of the SAAR through characterizing SAAR trends following transparent inter-facility comparisons. The pre-intervention period included SAARs from July 2017 through October 2017 and the post-intervention period included SAARs from November 2017 through June 2019. Antimicrobials were categorized according to NHSN definitions. Antibacterial agents used predominantly for hospital-onset infections (broad-spectrum agents) included anti-pseudomonal beta-lactams (including carbapenems) and intravenous aminoglycosides. Antibacterial agents predominantly used for resistant gram-positive infections included vancomycin, daptomycin, lipoglycopeptides, ceftaroline, oxazolidinones, and quinupristin/dalfopristin [9].

### 2.3. Motivational Tools 

Starting in October 2017, the SAAR was reported by the ASP at inter-professional system-wide antimicrobial subcommittee meetings. Physicians, infection preventionists, microbiology personnel, and pharmacists across the healthcare system were invited to attend. In addition to providing detailed SAARs for patient care locations and antimicrobial categories, the reports included inter-facility comparisons for the three hospitals in the system (hospitals A, B, and C) in order to highlight incongruities across the healthcare system and motivate key players of outlier patient care areas. Healthy competition among the hospitals was fostered. Pharmacists and physicians were encouraged to use the SAAR to inform targeted antimicrobial stewardship initiatives within their respective facility with ASP support. To further motivate key players, associations between unnecessary broad-spectrum agents and undesired clinical outcomes were highlighted, such as nephrotoxicity, *Clostridioides difficile* infection, and antimicrobial resistance using local data when available [18,19]. During subsequent system-wide meetings, the SAAR data were shared quarterly to provide updates and inform on the progress of targeted interventions.

### 2.4. Targeted Stewardship Interventions 

Using the pre-intervention SAAR report shared in October 2017, facilities enhanced the implementation of existing interventions to improve antimicrobial use with the goal of reducing the facility and specific patient-care location SAAR values. The targeted interventions involved electronic alerts to facilitate prospective audit and feedback for commonly used anti-pseudomonal beta-lactams (piperacillin/tazobactam, cefepime, and meropenem) and vancomycin in intensive care units (ICUs). These alerts fired upon a targeted antimicrobial order being active for >48 h. A locally derived clinical risk score for prediction of the probability of infections due to *Pseudomonas aeruginosa* was utilized to guide early de-escalation of anti-pseudomonal therapy in low-risk patients [20]. Clinicians were encouraged to use nasal methicillin-resistant *Staphylococcus aureus* (MRSA) polymerase chain reaction (PCR) to facilitate early discontinuation of anti-MRSA agents in hospitalized patients with pneumonia. Lastly, a focused initiative to improve the partnership and bidirectional education was deployed by daily (Monday through Friday) multidisciplinary rounds in the ICUs, including critical care specialists and antimicrobial stewardship pharmacists. All these existing interventions were simultaneously started in all three hospitals prior to the beginning of the study, as previously described [17]. Implementation of these interventions was further emphasized after sharing of the SAAR reports in October 2017.

### 2.5. Data Analysis 

A Student’s *t*-test was used to compare the mean SAAR in pre-intervention and post-intervention periods for all antimicrobials and specific patient care locations and antimicrobial categories within each hospital. JMP Pro (version 13.0, SAS Institute Inc., Cary, NC, USA) was used for statistical analysis. The level of significance for statistical testing was defined as 2-sided *p* < 0.05.

## 3. Results

The three hospitals maintained consistent participation in the NHSN antimicrobial use option during the 2-year study period. The mean pre-intervention SAARs for hospitals A, B, and C were 0.69, 1.09, and 0.60, respectively. Antimicrobial use of broad-spectrum agents predominantly used for hospital-onset infections and agents used for resistant gram-positive infections in the ICUs were particularly high in hospital B based on SAARs of 1.36 and 1.27, respectively (Table 1).

A plan was outlined to enhance ASP interventions targeting anti-pseudomonal beta-lactams and anti-MRSA agents in hospital B, particularly in the ICUs. Clinical leaders in various departments were involved to facilitate multidisciplinary collaboration. There was a significant decline in SAAR for overall antimicrobial use in hospital B from 1.09 in the pre-intervention period to 0.83 in the post-intervention period. SAAR for overall antimicrobial use in hospital B remained consistently less than 1 every month during the post-intervention period (Figure 1).

There were also significant declines in SAARs for broad-spectrum antimicrobial agents predominantly used for hospital-onset infections and agents used for resistant gram-positive infections in the ICUs in hospital B after the interventions (Table 1). The reductions in SAARs for these two categories were maintained during the 20-month post-intervention period (Figure 2 and Figure 3).

## 4. Discussion

In this multi-hospital healthcare system, reporting antimicrobial use to NHSN identified obvious opportunities for improvement in particular patient care locations and specific antimicrobial categories. Application of the SAAR as a motivational tool through transparent reporting and interfacility comparisons was followed by significant reductions in overall and broad-spectrum antimicrobial use in the ICUs in the outlier hospital. Encouraging healthy competition between multiple hospitals within a large healthcare system seemed to enhance the implementation of existing multi-faceted ASP interventions based primarily on local data. These findings support existing literature demonstrating the value of the SAAR as an antimicrobial stewardship metric for the identification of trends in antimicrobial use and opportunities for improvement [9,13,14,15,16].

The pre-intervention SAAR analysis revealed hospital B as an outlier within the healthcare system. The higher antimicrobial use in hospital B compared with hospitals A and C during the first 4 months of reporting to the NHSN antimicrobial use option was consistent with historic days of therapy data collected by the local ASP since January 2013 (data are not shown). Differences in antimicrobial use between the three hospitals within the healthcare system were previously attributed to variations in hospital characteristics and patient populations. However, the reality of excessive antimicrobial use in hospital B was more apparent when the SAAR data were presented at the system-wide ASP meeting in October 2017, as the SAAR metric is based on comparisons of each hospital’s antimicrobial use to its peer hospitals at the national level. This led to further investigation of the potential drivers of increased antimicrobial use. Through analysis of SAARs specific to antimicrobial categories and patient care locations, ICUs appeared to be the driver of increased antimicrobial use, specifically broad-spectrum agents predominantly used for hospital-onset infections and agents used predominantly for resistant gram-positive infections. Existing targeted interventions were emphasized across the healthcare system, including local tools to select appropriate empirical antimicrobial therapy and prospective audit and feedback for select antimicrobials. Periodic sharing of SAAR reports at system-wide antimicrobial stewardship meetings on a quarterly basis likely contributed to the sustained reduction of the SAAR in hospital B during the pot-intervention period.

Enrollment in the NHSN antimicrobial use option and access to SAAR data allow ASPs to incorporate a new antimicrobial stewardship metric into data reporting and tracking. Because the SAAR simplifies antimicrobial use into a ratio comparing actual to expected use, it is a digestible metric for inter-facility comparisons and communications with general healthcare providers who may not be familiar with traditional antimicrobial use metrics, such as days of therapy or defined daily doses. Literature is lacking to guide ASPs on what SAAR values should be targeted. As the current comparator is aggregated data from 2017, the local ASP aspires to achieve SAARs less than one for overall and broad-spectrum antimicrobials, assuming a level of maturation in nationwide ASPs over the previous three years with the inauguration of robust regulatory standards.

The integration of inter-facility comparisons within multi-faceted ASP interventions expands on the previous literature to suggest that the SAAR may have utility as a motivational tool in addition to its utility as an antimicrobial use metric [7,9]. The study has certain limitations. Concurrent reporting of inter-facility comparisons and implementation of multiple interventions limits the ability to ascertain the specific impact of each intervention. However, the confounding effect of multiple interventions was minimized by system-wide implementation of the interventions in all hospitals rather than targeting hospital B alone. A reduction in the SAARs for antimicrobial agents predominantly used for hospital-onset infections in the ICUs was observed in all three hospitals (Table 1). However, hospital B demonstrated the most drastic reduction in the SAAR for this antimicrobial category as well as significant reductions in overall antimicrobial use and agents predominantly used for resistant gram-positive infections in the ICUs. These observations may be an indicator of the motivational utility of SAAR reporting and inter-facility comparisons. The relatively short pre-intervention period represents another limitation of this study. Because of the perceived value of promptly sharing the SAAR data with the healthcare system, it was the desire of the ASP committee to distribute the information and initiate interventions with the availability of the data. In addition, antimicrobial use in the 4-month pre-intervention period was consistent with data previously analyzed across the health-system by the local ASP prior to participation in the NHSN antimicrobial use option. The study included SAAR data from three hospitals within one healthcare system. Future studies from larger healthcare networks would be useful to improve generalizability. The SAAR is a quantitative rather than qualitative antimicrobial stewardship metric. Future studies are needed to correlate the SAAR with quality of care and clinical outcomes. Finally, the SAAR is an antimicrobial stewardship metric that is currently available only to hospitals in the USA. An alternative method for adjustment of antimicrobial use by microbiological burden has been recently proposed and evaluated in a multicenter study in southeastern USA [9,21]. This novel antimicrobial stewardship metric for adjusted antimicrobial use by microbiological burden may be applied in other countries as well.

## 5. Conclusions

The utilization of SAAR reports allowed the identification of opportunities for improvement within specific locations and antimicrobial categories in the healthcare system. These findings support the motivational utility of the SAAR in reducing excessive antimicrobial use and alignment of the metric across multi-hospital healthcare systems. Transparent periodic SAAR reporting and inter-facility comparisons with the goal to foster competition enhance multi-faceted interventions and focus system-wide ASP efforts.

## Figures and Tables

**Figure 1 pharmacy-09-00032-f001:**
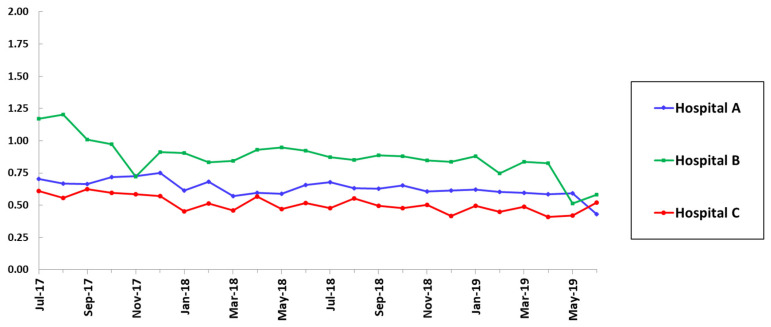
Standardized antimicrobial administration ratio (SAAR) trends for all antimicrobials used in adult intensive care units (ICUs), wards, step down units, and oncology units.

**Figure 2 pharmacy-09-00032-f002:**
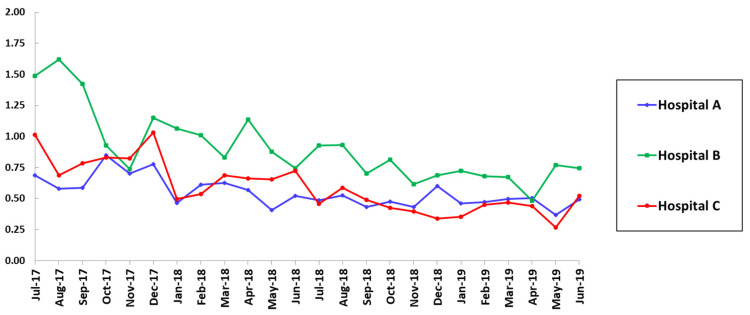
SAAR trends for broad-spectrum agents used for hospital-onset infections in adult ICUs.

**Figure 3 pharmacy-09-00032-f003:**
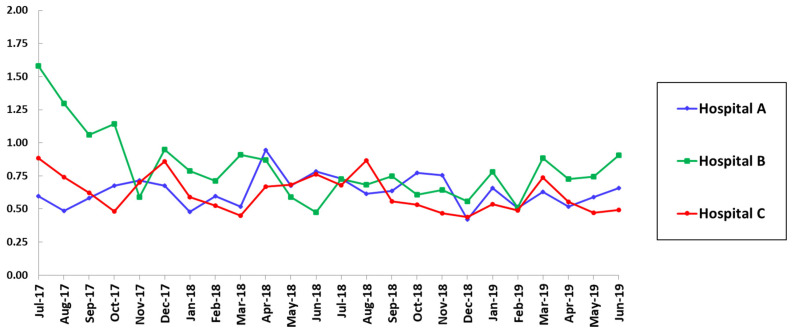
SAAR trends for antimicrobial agents used for resistant gram-positive infections in adult ICUs.

**Table 1 pharmacy-09-00032-t001:** Pre- and post-intervention mean standardized antimicrobial administration ratio (SAAR) for antimicrobial categories and patient care locations. ICU, intensive care unit.

SAAR Category	Hospital A			Hospital B			Hospital C		
	Pre	Post	*p*-Value	Pre	Post	*p*-Value	Pre	Post	*p*-Value
All agents, all locations	0.69	0.62	0.06	1.09	0.83	<0.001	0.60	0.64	0.69
Broad-spectrum agents used for hospital-onset infections, ICU	0.67	0.52	0.01	1.36	0.81	<0.001	0.83	0.54	0.007
Agents used for resistant gram-positive infections, ICU	0.59	0.64	0.37	1.27	0.72	<0.001	0.68	0.60	0.31

## Data Availability

The data presented in this study are available on request from the corresponding author. The data are not publicly available due to institutional policy.

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
