# Peer review of "Application of Standardized Antimicrobial Administration Ratio as a Motivational Tool within a Multi-Hospital Healthcare System"

_pharmacy, 2021, doi:10.3390/pharmacy9010032_

Round 1

Reviewer 1 Report

Dear Authors,

Excellent study, clean and very well presented, however several clarifications should be included in order to improve and make it useful for all countries to fight AR. 

Here my comments/questions 
- Regarding the target or category. Do you have data about SAAR in a pediatric cohort? What about gender?
- Hospital A and C according to the graphics don’t show a significant change since SAAR application, what are they doing differently from 2017?

- Is the number of patients different in Prisma Health–Midlands? It is comparable SAARS in all 3 hospitals?
- What is the main limitation of this study? Could it be applied in all countries? Middle-low? Is there any software or platform where clinicians and personal involved gather all this information with accuracy and reliability?
If gram-negative bacteria are the most critical in nosocomial infections, why didn’t you include data about the treatment? I think the graphics could change drastically. The study is based just on broad-spectrum and gram-positive bacteria.
Thank you for these clarifications
Kind regards

Author Response

Dear reviewer,

Thank you very much for your kind words and thoughtful review. Please see responses to your questions and comments below.

1- Regarding the target or category. Do you have data about SAAR in a pediatric cohort? What about gender?

The 3 facilities included in this study treat adult patients. We have a pediatric hospital associated with our healthcare system which was not included in the study since it has a separate antimicrobial stewardship team and different interventions. The SAAR reports provided by NHSN include specific antibiotic use data for antibiotic use categories and patient care locations, but not for gender or age.

2- Hospital A and C according to the graphics don’t show a significant change since SAAR application, what are they doing differently from 2017?

We believe the significant reduction in overall SAAR for hospital B compared to hospitals A and C highlight the potential motivational application of performing inter-facility comparisons. Similar interventions were implemented across the 3 hospitals and remain in place to date. We believe this suggests that highlighting hospital B as an outlier with the original SAAR report led to an alignment of their antibiotic use with Hospital A and C through motivating competition in addition to the interventions which were implemented.

3- Is the number of patients different in Prisma Health–Midlands? It is comparable SAARS in all 3 hospitals?

The 3 hospitals included varied in size. We added a sentence to the Methods section to clarify that the three hospitals varied in bed capacity and patients’ characteristics without revealing the specific number of licensed beds at each hospital since that would lose the anonymity. We believe that the SAAR should be comparable across the three hospitals since each one is compared to their peer hospitals in this national metric.

4- What is the main limitation of this study? Could it be applied in all countries? Middle-low? Is there any software or platform where clinicians and personal involved gather all this information with accuracy and reliability?

The main limitation of this study was the small number of included hospitals. Future studies from larger healthcare networks would be useful to improve generalizability. This limitation was added to the Discussion section.

Participation in the NHSN’s AU Option is currently limited to facilities within the United States, so this specific benchmarking model applies only to hospitals in the United States. However, there is an alternative method for adjusting antimicrobial use by microbiological burden that may be applied in other countries as well. This statement and appropriate citations were added to the Discussion section.

5- If gram-negative bacteria are the most critical in nosocomial infections, why didn’t you include data about the treatment? I think the graphics could change drastically. The study is based just on broad-spectrum and gram-positive bacteria.

Thank you for this comment. Throughout the manuscript, the reference to broad-spectrum antibacterial agents refers to the NHSN antimicrobial use category of “Broad-spectrum antibacterial agents used predominantly for hospital-onset infections” (lines 95-97). This category captures data on nosocomial infections based on the antibacterial agents included in it. We have updated the manuscript to clarify.

Reviewer 2 Report

I congratulate the authors on a well-thought out study and excellent presentation of the results.

115-130: Were all the mentions targeted interventions implemented in all 3 hospitals? And when did these interventions start? Likely at the start of the post-intervention period?

128-129: How often did the multidisciplinary rounds occur?

It is unclear when the targeted interventions were implemented and I would include this in the manuscript. 

If the interventions were implemented in conjunction with the report of each facility's SAAR- it will be unclear whether the improvement in hospital B's SAAR is directly related to the SAAR report, implementation of the targeted stewardship interventions, or both. I would include this in the limitation. 

Author Response

Dear Reviewer,

Thank you for your kind words and thoughtful review. Please see responses to your questions and comments below.

1- 115-130: Were all the mentions targeted interventions implemented in all 3 hospitals? And when did these interventions start? Likely at the start of the post-intervention period?

All the targeted interventions were simultaneously started in all three hospitals prior to the beginning of the study as previously described in reference #17. Implementation of these interventions was further emphasized after sharing of the SAAR reports in October 2017. We added a sentence in this regard to the Methods section to clarify.

2- 128-129: How often did the multidisciplinary rounds occur?

Multi-disciplinary rounds occurred daily (Monday through Friday). The manuscript has been updated to reflect this. (Line 128)

3- It is unclear when the targeted interventions were implemented and I would include this in the manuscript. If the interventions were implemented in conjunction with the report of each facility's SAAR- it will be unclear whether the improvement in hospital B's SAAR is directly related to the SAAR report, implementation of the targeted stewardship interventions, or both. I would include this in the limitation.

Thank you for this comment. The targeted interventions were already in place prior to the beginning of the study in July 2017. However, implementation of these interventions was emphasized after sharing the SAAR reports in October 2017.

This manuscript is a resubmission of an earlier submission. The following is a list of the peer review reports and author responses from that submission.